# Detection of Mutual Exciting Structure in Stock Price Trend Dynamics

**DOI:** 10.3390/e23111411

**Published:** 2021-10-27

**Authors:** Shangzhe Li, Xin Jiang, Junran Wu, Lin Tong, Ke Xu

**Affiliations:** 1School of Mathematical Science and LMIB, Beihang University, Beijing 100191, China; by1809107@buaa.edu.cn; 2School of Computer Science and Engineering and NLSDE, Beihang University, Beijing 100191, China; wu_junran@buaa.edu.cn (J.W.); kexu@buaa.edu.cn (K.X.); 3Zhengzhou Aerotropolis Institute of Artificial Intelligence, Zhengzhou 451162, China; 4Pengcheng Laboratory, Shenzhen 518052, China; 5Gabelli School of Business, Fordham University, New York, NY 10023, USA; ltong2@fordham.edu

**Keywords:** self- and mutually exciting processes, Hawkes process, stock price trend dynamics

## Abstract

We investigated a comprehensive analysis of the mutual exciting mechanism for the dynamic of stock price trends. A multi-dimensional Hawkes-model-based approach was proposed to capture the mutual exciting activities, which take the form of point processes induced by dual moving average crossovers. We first performed statistical measurements for the crossover event sequence, introducing the distribution of the inter-event times of dual moving average crossovers and the correlations of local variation (LV), which is often used in spike train analysis. It was demonstrated that the crossover dynamics in most stock sectors are generally more regular than a standard Poisson process, and the correlation between variations is ubiquitous. In this sense, the proposed model allowed us to identify some asymmetric cross-excitations, and a mutually exciting structure of stock sectors could be characterized by mutual excitation correlations obtained from the kernel matrix of our model. Using simulations, we were able to substantiate that a burst of the dual moving average crossovers in one sector increases the intensity of burst both in the same sector (self-excitation) as well as in other sectors (cross-excitation), generating episodes of highly clustered burst across the market. Furthermore, based on our finding, an algorithmic pair trading strategy was developed and backtesting results on real market data showed that the mutual excitation mechanism might be profitable for stock trading.

## 1. Introduction

In numerous complex systems, activity is driven by endogenous and exogenous factors. The endogenous factors mainly refer to intrinsic self-excitation mechanisms, while the exogenous factors come from environmental changes or other events that take place independently of events that have occurred in a system. Typical instances could be found in seismic dynamics [1,2,3,4], tweets in social networks [5], neuronal firing [6,7], and so on. In this respect, financial markets are the most representative. The movement of asset prices is consistently affected by the occurrence of internal events, such as large orders and abnormal price changes related to past events [8,9,10,11,12], and also by the arrival of exogenous events, such as news and economic policies [13,14]. To identify the endogenous and exogenous effects and characterize how they influence movements in financial markets has been a crucial problem for decades.

Hawkes processes [2,15,16], described as point processes with linear or nonlinear interactions, are frequently used to model non-stationary dynamics of financial activities driven by endogenous and exogenous joint factors [17,18]. In Bowsher’s pioneering work [19], a bivariate Hawkes process was introduced in order to model the joint dynamics of trades and mid-price changes of the NYSE. Since then, a great number of works have been proposed, which include market activity or risk models [9,19,20,21,22], price models [10,11,23], and impact models [9,23,24]. The most attractive feature of the Hawkes model is that the intensity consists of two terms: one term is a background rate, which accounts for the environmental exogenous effect, and the second term stands for the triggering effect from the preceding events, which could represent the endogenous self-excitation or other kinds of combination excitation [13].

In a recent series of articles, several constructive works [13,22,25,26,27,28] have addressed the important problem of a mutually exciting mechanism in stock markets within the framework of Hawkes models. In [27], Ait-Sahalia et al. proposed an effective mutually exciting model to capture the jump-diffusion process in global stock markets. It is well demonstrated that a jump in one region of the world or one segment of the market increases the intensity of jumps occurring both in the same region (self-excitation) as well as in other regions (cross-excitation). Actually, this mutual excitation induces a special kind of correlations between markets or different segments of one market. We call such correlations mutual excitation correlations in the following discussion.

In this study, we proposed a multidimensional Hawkes-model-based approach for the characterization of mutual excitation correlations in stock markets. Instead of checking micro activity such as jumps or transactions in price change, we considered events identified by technical trading indicators [29,30]. The profitability of technical trading indicators has been discussed widely in the technical analysis literature [30,31,32,33,34]. The majority of trading rules based on technical indicators stem from the assumption that prices move in trends determined by the changing attitudes of traders towards various economic, political, and psychological forces. In this sense, the activity or patterns induced by technical indicators form a mesoscopic portrayal of market price movements.

Specifically, we studied the dynamics of crossovers of moving average lines in stock market. Moving average crossover is one of the most popular trend-determining techniques based on the crossing of two moving averages of prices. This technique is also known as the dual moving average crossover (henceforth dual moving average crossovers) rule [29] in the technical analysis literature. Dual moving average crossovers are considered to be a good indicator of a market movement since the “microstructure” noise could be partially eliminated by price averaging. To the best of our knowledge, there are few works using mesoscopic economic activity such as dual moving average crossovers of prices to study the inner relevance in stock markets. We applied our approach to the analysis of stock price data from Chinese A-share markets. In [28], Hung T. Diep and Gabriel Desgranges present some very interesting results about the dynamics of the price behavior in stock markets, such as that an assembly of agents in a financial market interact with each other, which involved some exogenous factors like the economic temperature as well as economic measures taken by the government. We also identified a large number of golden and death crossovers in all the stock sectors. Interestingly, our approach demonstrates that the crossover arrival dynamics in most stock sectors are generally more frequent than the standard Poisson process.

Our main original contribution is a Hawkes model of the crossover activity in various stock sectors where a multidimensional kernel describes the mutual effect among different stock sectors. We found that a triggering structure of stock sectors could be revealed by the mutual excitation correlations obtained from the excitement kernel matrix of our model. It is significant to emphasize that mutual excitation correlations can be applied to design a profitable trading strategy since the crossover activity studied in our model is fundamental for the classical moving average (MA) trading rules [29]. To make a preliminary attempt, we proposed a quantitative pair trading strategy verified by back-testing on real stock market data.

The remainder of the article is organized as follows. In Section 2, we present our data set of prices and the main variables introduced to characterize the MA. We then perform statistical measurements for the crossover event sequence like the method used in spike train analysis. Meanwhile, we introduce the model of the univariate Hawkes processes and multidimensional Hawkes processes. In Section 3, we show how Hawkes models are able to describe the dynamics of an MA cluster in sectors when only the endogenous component is considered, and then we choose the leading sector and present our modeling approach, where a new component of the Hawkes process is added, taking into account the effect of the leading sector. Moreover, we present an empirical performance comparison of trading portfolios, which use our finding with the buy-and-hold strategy on the sector index. Finally, we present some conclusions Section 4.

## 2. Materials and Methods

### 2.1. DATA

We used daily frequency price data of more than 2000 stocks in Chinese A-share markets, purchased from the Wind data service. Specifically, these stocks were divided into 10 sectors according to the Global Industry Standard Classification (GISC), which included energy, materials, consumer discretionary, consumer staples, health care, financials, information technology, telecommunication services, utilities, and industrials. Details of each stock sector are demonstrated in Table 1. Here, the telecommunication sector was not taken into consideration due to a very few number of stocks. The timestamps of the data set were restricted to the regular session (9:30–11:30 and 13:00–15:00) from 4 January 2016 to 31 December 2018. For convenience, in the following discussion the time scales are normalized into interval (0,100).

The change of stock prices is full of random noise and cannot serve as a good indicator of market movement. Thus we alternatively choose the crossovers of moving average lines as filtered price transactions and analyzed the time series of the crossover events. Basically, moving averages are considerable parts in the ensemble of tools in technical analysis trading. The properties and efficacy have been investigated in many previous academic studies. Without loss of generality, here we considered a commonly used moving average indicator, the exponential moving average (EMA). Denoting the closing price $P_{n}\left(n\in N_{+}\right)$ of a stock at time *n* and denoting $E_{n}\left(k\right)$ as the $k^{th}$ day backward exponential moving average at time *n*, the calculation formula reads:(1)$$E_{n}\left(k\right)=2\left(P_{n}-E_{n-1}\left(k\right)\right)/n+E_{n-1}\left(k\right),$$
where a large (small) value of parameter *k* indicates a long-term (short-term) moving average. To filter the price change with crossovers, we considered the long-short EMA cross-over, which also performs as a basic trading strategy. Concretely, a “buy” signal could be issued when the shorter EMA crosses above the longer EMA, which is also named as a “golden crossover.” Similarly, “sell” signals can be defined in the opposite direction, which is called a “death crossover” as the longer EMA crosses above the shorter one. This technique is also known as the dual moving average crossover (henceforth dual moving average crossovers) rule [29] in the technical analysis literature. Technical analysts argue that when price moves upwards, the shorter moving average will rise faster than the long-term one, which is less responsive to recent price changes, indicating buying pressure and the possibility of a future bullish trend. Thus, for investors, the appearance of a golden crossover indicates that one should open a long position and maintain the position until the short-term average cross below the long-term one. By applying the same reasoning, investors should sell the asset when a death crossover appears.

In this study, we were concerned with the statistical properties of golden and death crossover signals, which can be represented as binary variables. The golden crossover signal variable $S_{n}^{G}$ at time *n* based on an EMA cross-over is defined as follows:(2)$$S_{n}^{G}=\left\{\begin{array}{cc} 1, & \;E_{n-1}\left(k_{s}\right)<E_{n-1}\left(k_{l}\right)\&E_{n}\left(k_{s}\right)>E_{n}\left(k_{l}\right),\\ 0, & \;else. \end{array}\right.$$

In the same manner, the death crossover signal $S_{n}^{D}$ at time *n* can be obtained as: (3)$$S_{n}^{D}=\left\{\begin{array}{cc} 1, & \;E_{n-1}\left(k_{s}\right)>E_{n-1}\left(k_{l}\right)\&E_{n}\left(k_{s}\right)<E_{n}\left(k_{l}\right),\\ 0, & \;else. \end{array}\right.$$

Here, $k_{s}$ and $k_{l}$ are short-term and long-term parameters, respectively. In the following study, we took $k_{s}=12$ and $k_{l}=26$ as general settings. To make an intuitive description, in Figure 1 we demonstrate an EMA golden crossover and a death crossover of a selected stock. Some statistical results of the stock data, such as numbers of stocks and crossovers in each sector, are presented in Table 1.

### 2.2. Statistical Measure for Crossover Event Sequences

In order to investigate events with a more distinct financial meaning, we mainly focused on those FIRST crossover events of each stock instead of all the crossover events. The first moving average crossover in a given time window has significant behavioral finance meanings, which indicates a potential beginning of a trend among the stocks in a certain sector. We were more concerned about the excitation between different sectors in a given time window. The first crossover event could be a nice representation of the trend dynamics of one sector. It is also easier to identify these first crossovers so as to better distinguish the excitation effects between different sectors. In this sense, the first EMA crossovers of each stock in a certain sector during a given period were selected. We note here that since the data granularity was 1 day, to distinguish those events happening on the same day, we randomized the corresponding event times by adding each timestamp expressed in days with a random number uniformly distributed in (0,1]. In Figure 2 we illustrate how those first golden EMA crossovers in the financial and IT sector were distributed.

To study the statistical proprieties of the EMA crossover, we first considered inter-event times, which indicate the time interval between each pair of events [35]. Let us consider a sequence of interevent times $T_{1},T_{2},\cdots ,T_{n}$ between n+1 golden (death) crossover events, and $T_{i}$ represents the length of the ith time interval. We carefully checked the distribution of $T_{i}$ in various stock sectors, and it was demonstrated that the inter-event times between successive events in any chosen stock sector distribute as a power-law form, which is significantly divergent from the exponential distributions expected in a classical Poisson process. In Figure 2, we show an example, which is the distribution of $T_{i}$ in the financial sector. In Table 2, we show the power-law constants of the inter-event times distributions for each sector. The larger value of the power-law constant in the CS sector may indicate a weak correlation between the crossover activity of intra-stocks, while, in the financial sector, the power-law constant was smaller, which indicates a more active trend dynamic pattern. We can use the variances of these distributions to adopt different investment strategies for different sectors. Therefore, it was necessary to study the excitation effect between different sectors.

This observation motivated us that the dynamics of crossover events might not be regular. In fact, similar questions are also investigated in computational neuroscience, where neuronal activities called spike trains are extensively studied [36]. Revelation of inner connectivity or relations from various patterns of event actions have been widely attempted for neural circuits or online social contacts [37,38]. To make a quantitative description, we analogically introduced a promising measure, which was originally introduced in spike-train data analysis [38,39], the local variation(LV). It has been verified that the LV measure is quite efficient and robust in characterizing whether the activity rate of a event sequence is regular or intermittent [38]. Generally, the LV is defined as follows:(4)$$LV=\frac{1}{\left(n-1\right)}\sum_{i=1}^{n-1}\frac{3{\left(T_{i}-T_{i+1}\right)}^{2}}{{\left(T_{i}+T_{i+1}\right)}^{2}}$$

We note here that the factor 3 in Equation (Equation 4) was to make the LV value for a standard Poisson process equal to unity. In this sense, if LV<1, the inter-event times are smaller than those expected in a Poisson process, which means the event sequence is more regular than a Poission one. Otherwise, if LV>1, the event sequence is much more intermittent. To make a more comprehensive sketch of the crossover event sequence, we divided the whole time interval into *N* times windows and checked the corresponding LV values for each time window. In this way, we obtained a sequence of LV as $LV_{1}^{i},LV_{2}^{i},\cdots ,LV_{N}^{i}$ for each stock sector *i*.

An intuitive question is whether these LV sequences for different stock sectors are correlated. Since the answer to this question may help reveal the underlying connection structure of the stock market, we made a direct attempt by checking the Pearson correlation coefficient between two LV sequences. Take the IT and financial sectors for instance. We checked the distributions of corresponding LV sequences in the two sectors and illustrated the Pearson coefficient in a contour manner, as shown in Figure 3.

For the whole stock market, the correlation coefficients for each pair of stock sectors were examined. For both golden crossover and death crossover events in 9 stock sectors, we obtained two 9×9 coefficient matrix accordingly in Figure 4.

In this study, we tried to answer three questions. Does the activity rate of crossover events show regular or intermittent features? What is the relationship or interconnection between the event sequences for different stock sectors? Is there a possible mathematical model that could characterize the inner structure of the crossover event dynamics?

### 2.3. The Univariate Hawkes Processes

Actually, the arrival of EMA crossovers can be viewed as discontinuous random processes. Numerous kinds of models have been proposed to investigate the discrete nature of market pattern arrivals. The Poisson point process is a fundamental case in which events arrive independently of one another with a constant arrival rate λ. However, due to a lack of correlation structure, the Poisson point process can hardly capture the stylized features of markets, such as clustering of stock EMA crossover arrivals in some sectors. In this sense, the self-excited Hawkes model is a more consistent approach, which can be regarded as the generalization of the nonhomogeneous Poisson process, where the intensity $\lambda_{t}$ of arrival events at time *t* depends on the history of the process $H_{t}$ according to:(5)$$\begin{array}{c} \lambda \left(t|H_{t}\right)=\mu \left(t\right)+\sum_{t_{i}<t}\phi \left(t-t_{i}\right), \end{array}$$
where the first term μ(t) is a background rate, which accounts for exogenous events that are independent of history, and the second term represents the triggering effect from the preceding events at the occurrence time $t_{i}$, the strength of which is controlled by a triggering kernel function ϕ(·).

For our purposes, we first simply introduced the univariate Hawkes process approach to study the EMA crossover arrivals of stocks within a single sector. We assumed that the background (triggering) term was exclusively attributed to the exogenous (endogenous) effect. From another point of view, this univariate Hawkes process can be mapped into the well-known branching processes. The strength of the endogenity of the process is then characterized by the branching ratio $\int_{0}^{\infty }\phi \left(s\right)ds$, that is, the average number of events that are directly triggered by a single event. The branching ratio is also interpreted as a ratio of the events with endogenous origin to all the events. Therefore, exogenous “mother” events occur with intensity μ(t) and give rise to endogenous “child” events, which themselves give birth to more child events.

Via estimation of Hawkes processes parameters, the functional form of the kernel ϕ(·) is essential. In empirical studies, one can choose an *a priori* functional form for the kernel, depending on some parameters, and then determine the parameters by maximizing the likelihood over the observed data. In our application, for the endogenous kernel ϕ(·), we adopted an exponential kernel that does not resort to rejection sampling. This kernel was proposed by Dassios and Zhao [2013] and scales linearly to the number of events drawn.
(6)$$\begin{array}{c} \phi \left(s\right)=\alpha e^{-\beta s}, \end{array}$$
where α and β are parameters. Especially, the parameter α corresponds to the branching ratio (magnitude), while β fixes the time scale. It is worth to note here that the power law function form is also common for the triggering kernel in the context of quantitative finance, especially in accounting for the long-range memory. In this study, since we only considered EMA crossovers of some stocks in a fixed short interval, which is lacking in long-range memory, we chose the exponential form for computational simplicity.

### 2.4. The Mutual and Multidimensional Hawkes Processes

At this step, we present the main contribution of this study: a potential sector–sector leading impact mechanism recovered by a multidimensional Hawkes process, which characterizes the mutual effects between each pair of stock sectors. Actually, this model was designed to figure out the impact of EMA crossover arrivals within one sector on stock activities (EMA crossovers) in another sector. Namely, the crossover activities of one sector were now not only a self-excited point process but were also influenced by other sectors. Concretely, when considering a selected stock sector $A_{1}$, the crossover events in $A_{1}$ forms the points process, and we assumed that the arrivals of crossover events in other sectors $A_{2},A_{3},A_{4},\cdots ,A_{M}$ forms the external events that contribute additional exciting intensities to the process in $A_{1}$. In this sense, the crossover process outside sector $A_{1}$ was considered to be deterministic in the mutual model. Since we aimed at finding out the mutual impact among the sectors, it was reasonable to regard the events in other sectors that could significantly affect the evolution of the selected sector. Generally, the effect of such external events could be modeled via a separate term in the intensity expression. For λ(t) in some given sector *i*, the mutual model can be expressed as:(7)$$\begin{array}{c} \lambda^{i}\left(t\right)=\mu^{i}+\sum_{t_{i}<t}\phi^{i}\left(t-t_{i}\right)+\sum_{j=1}^{M-1}\sum_{t_{k}^{j}<t}\phi^{j}\left(t-t_{k}^{j}\right), \end{array}$$
where $\mu^{i}$ is still the exogenous intensity, and $\phi^{i}\left(\cdot \right)$ is the endogenous self-exciting kernel for sector *i*. The third term, $\phi^{j}\left(\cdot \right)$, is the exogenous kernel that accounts for the external events’ excitements from sector *j*, which were assumed to be deterministic and known in advance. As we said, these special exogenous events increase the rate of immigrants’ arrival via memory kernels $\phi^{j}$, which, in general, could be distinguishable both in terms of the magnitude and the relaxation for each *j* via using the exponential form. In this way, both $\phi^{i}\left(\cdot \right)$ and $\phi^{j}\left(\cdot \right)$ were regarded as excitement terms.

Analogously, for each sector *i*, one can obtain a similar mutual equation like Equation (Equation 7). Considering there was a total of *M* sectors, an M-dimensional Hawkes point process $N\left(t\right)=(N^{i}\left(t\right)),i\in \left[1,M\right]$ for every *i* can be defined. By combining both the internal and external terms, the i-th component of its intensity function $\lambda^{i}\left(t\right)$ is a linear regression of the past jumps of N(t), i.e.,
(8)$$\begin{array}{c} \lambda^{i}\left(t\right)=\mu^{i}+\sum_{j=1}^{M}\sum_{t_{k}^{j}<t}\phi_{E}^{ij}\left(t-t_{k}^{j}\right), \end{array}$$
where $\mu^{i}$ is the exogenous intensity and $\phi_{E}=\phi_{E}^{ij}\left(\cdot \right),1\leq i,j\leq M$ is the Hawkes excitement kernel matrix, where each $\phi_{E}^{ij}\left(\cdot \right)$ is a real positive function. Alternatively, the matrix convolution form of Equation (Equation 9) simply rewrites:(9)$$\begin{array}{c} \lambda \left(t\right)=\mu +\phi_{E}*dN_{s}. \end{array}$$

As described in [11,25], the stability condition of this multidimensional Hawkes process is decided by the $L_{1}$ norm matrix of kernel $\phi_{E}$. Namely, if the $L_{1}$ norm matrix,
(10)$$\begin{array}{c} \left|\right|\phi_{L}\left|\right|=\left(|\right|\phi_{E}^{ij}{\left|\right|}_{1}),1\leq i,j\leq M, \end{array}$$
has a spectral radius strictly smaller than one, then the process $N_{t}$ admits a version with stationary increments.

Similarly, we also specified the functional form of $\phi_{E}^{ij}$ as the exponential kernel. As we discussed in the univariate part above, the exponential kernel gives good results for the excitement component, that is,
(11)$$\begin{array}{c} \phi_{E}^{ij}=\alpha^{ij}e^{-\beta^{ij}t}. \end{array}$$

We note here that the norm of $\phi_{E}^{ij}$ indicates the mean number of events of type *i* triggered by an event of type *j*, which can be used as as an indicator for estimating the impacts from sector *j* working on sector *i*.

## 3. Results and Discussion

### 3.1. Single Sector

We estimated the best parameters for each stock sector separately by maximizing the log likelihood. The whole time interval was divided into *N* time windows of equal length *w*. As we mentioned in the data section before, time stamps were normalized into (0,100), and w∗N=100 should be satisfied. Further, since there was a total of three-year daily data, the time window also had a corresponding real time length meaning, i.e., if the time window is designed to represent three months, then *w* should be set as 100/12 since there are twelve three-month time windows in three years. In this way, we could obtain a set of optimal parameters for each sector with a given time window length *w* as in Table 3. Basically, the log-likelihood function of the univariate Hawkes process of given parameters θ based on the data $D^{\left[S,T\right]}={\left\{t_{i}\right\}}_{i=1}^{n}$ of *n* events in an observation interval [S,T] is given as
(12)$$\begin{array}{c} logL\left(\theta |D^{\left[S,T\right]}\right)=\sum_{i=1}^{n}log\lambda \left(t_{i}|H_{t}\right)-\int_{S}^{T}\lambda \left(t|H_{t}\right)dt \end{array}$$
where log represents the natural logarithm throughout the article. Further, it was necessary to determine the best time window length *w* for studying the first-crossover events. In fact, the likelihood values can be considered as a score for ranking the performance of each *w*. Higher scores correspond to a better fitting effect to the Hawkes model. In this way, various likelihood values corresponding to various values of *w* ranging from 1 to 10 were calculated through the estimation process. Results for different stock sectors indicate that, for most sectors, w=2 or w=3 should be a favorable choice, which is shown in Figure 5. In the following discussion, we prefered to set w=3, since three month is also one quarter in the financial cycle, which has distinct economical meanings.

Overall, we argue that the univariate Hawkes model with the exponential kernel provides a nice description of the empirical data for individual stock sectors. Based on the conclusion in [13], the Hawkes process is stationary if
(13)$$\begin{array}{c} n=\int_{0}^{\infty }\phi \left(\tau \right)d\tau <1. \end{array}$$

Here, the quantity *n* is the fraction of the average intensity induced by the self-exciting part. As *n* became close to 1, most of the activity could be regarded to be induced by the underlying self-exciting factors, whether they hence appeared to be endogenously generated.

### 3.2. Multiple Sectors

Similar to Equation (Equation 12), the log likelihood function of our multidimensional model can also be analytically derived. In this sense, we could estimate a set of parameters in Equation (Equation 11). Concretely, for each sector in the data set, the multidimensional model was calibrated referring to a time window of length w=3 as we discussed in the univariate case. First, the exogenous intensity $\mu^{i}$ of each sector was investigated through the multidimensional model. In Table 4, we show the $\mu^{i}$ values together with the average number (activity) of golden and death crossover events in each sector. Compared with the $\mu^{i}$ values obtained in the univariate model (refer to Table 1), the updated exogenous intensity in the multidimensional model appeared to be much smaller for each sector. We argue that this means the crossover dynamics may be dominated by the triggering term, which is also consistent with our hypothesis.

The set of parameter pairs $\left(\alpha^{ij},\beta^{ij}\right)$ can be estimated by executing regression of the multidimensional model on the real data set. After that, one could gain the norm matrix $\left|\right|\phi_{L}\left|\right|$ defined in Equation (Equation 10), in which the matrix element $\phi_{E}^{ij}$ indicates influence divergences for individual sectors. As illustrated in Figure 6, we present the norm matrix $\left|\right|\phi_{L}\left|\right|$ for both golden and death crossover cases in the heat map manner. The triggering impact of sector *i* indexed in the horizontal axis and on sector *j* indexed on the vertical axis is identified by distinguishable colors. This triggering impact matrix actually provides a landscape of the price-trend-inducing relationship between nine stock sectors in the Chinese A-share market. For instance, in the golden crossover heat-map, the financial sector, in most cases, had a stronger triggering effect on other sectors, which means the golden crossover of financial stocks might, in large probability, cause several cascading golden crossovers in other sectors. While, in the death crossover case, the energy sectors seemed to play a leading fall-down role, that is, the death crossovers of energy stocks may possibly induce price down crossovers in other sectors. Figure 6 also shows that the matrix is asymmetric, which reveals an additional correlation structure than the Pearson correlation coefficient matrix presented in Figure 4. It is interesting to see that although the financial sector had a strong impact on other sectors, it was not strongly self-excited in the golden crossover case, while, for the death crossover, the IT sector showed a considerable self-excited signal among all the sectors, which is also consistent with the exogenous intensity μ=0.686 (largest) of IT calculated in Table 4. This indicates a cascading fall down of price is likely to happen within the IT sector. In Table 5, the average impact of each sector on the other ones is detailed.

Additionally, the condition for stationarity of Equation (Equation 9) was also verified by checking the eigenvalues of the norm matrix $\left|\right|\phi_{L}\left|\right|$. For both the golden and death crossover cases, the eigenvalues of the corresponding matrix were presented in the complex plane. It was demonstrated that all the eigenvalues were located within the unit circle on the complex plane, as illustrated in Figure 7, which ensured the stationarity of the data.

### 3.3. Strategy

As previously outlined, one of the motivations for this study was to build a profitable trading strategy based on the triggering structure revealed by our multidimensional model. To make a preliminary attempt, we utilized the triggering impact of sector *A* (leader sector) on sector *B* (audience sector) to generate buy and sell signals on sector *B* during stock trading. From the golden crossover heat-map, it can be seen that the financial sector has a considerable triggering impact on the material sector; thus, we took the financial sector as the leader one and the material sector as the audience one. The trading period was the same as the previous one used above, i.e., due to regulations in the Chinese stock market, one can only execute long operations on the stocks. The trading signals can be generated as follows: for the buying signal, once the number of golden EMA crossovers of the leading sector within one day exceeds a certain threshold $\Delta_{B}$, we will, before the close of the day, buy all the stocks that have not encountered their first golden EMA crossovers. Similarly, for the long selling signal, if the number of death EMA crossovers of stocks in the leader sector exceeds a certain threshold $\Delta_{S}$, we will sell the holding stocks that have not encountered their first death EMA crossover at the close of the day. To avoid fluctuation risk, another technical indicator, moving average convergence divergence (MACD), was also used. If the MACD value of the stock becomes smaller than 0, the long selling operation is also executed. A detailed list of trade rules can be found in Table 6. To avoid fluctuation risk, one common technical indicator, MACD [30], was also introduced here. If the MACD value of the stock becomes smaller than 0, the long selling operation is also executed. Without loss of generality, we numerically tested various values from 2015 to 2018 and finally set the threshold $\Delta_{S}=\Delta_{B}=4$ to achieve the maximal profit. Typical transaction costs for a round trip (purchase and sale) of a stock were set as 0.05%, which is the same in the real market trading case. In order to better verify the results we found, we not only conducted the backtesting from 2015 to 2018, but we also conducted the backtesting in 2019 and 2020 with the same strategy and parameter settings for 2015–2018.

The returns for the trading strategy described above are illustrated in Figure 8. The result of net value revealed that this strategy can generate significant profits during the six-year evaluation period. To evaluate trader performance, we compared the daily portfolio value of our strategy with the buy-and-hold strategy on the index of the audience sector in our six-year evaluation period.

Figure 8 shows the daily return of the two strategies over the assessment period. The daily earnings were calculated based on the earning performance of the previous trading day. Compared with the change of −6.5% in the material index, our strategy yielded a remarkable return of 188.80% in 2015–2018, and our strategy also achieved fantastic returns in 2019–2020. If we hedge with the index of the material sector, the net curve can be smoother and higher. With reasonable conditions, the proposed quantitative pair trading strategy is quite promising to find the leader relationship of the two sectors.

## 4. Conclusions

In this study, we introduced a Hawkes-process approach to the description of the novel leading relationship between nine sectors in Chinese A-share market, in line with the first crossover of the exponential moving average of the close price in a fixed interval. The leading relationship is very important and might reveal the sensitivity of sectors to the market activity. We first performed statistical measurements for a crossover event sequence that was similar to a spike train in neuronal activities and then considered the unconditional Hawkes modeling, i.e., we only considered a single sector. We found that the criticality parameters seem to be pretty insensitive to the change of the tick size, while the parameters of the kernel showed an abrupt change around it, and we obtained the best length of interval by comparing the score of each sector. We then considered our main original contribution, namely, the introduction of a new leading relationship between nine sectors in the Chinese A-share market. First, we chose the leading sector by the matrix of the norms of the kernel. Then, in both cases, we showed that the model with the leading sector kernel(s) outperformed the simple Hawkes model with only the endogenous kernel. We also noted that, once the leading sector term was introduced, the model obtained a better fitting. This suggests that, in the presence of localized exogenous excitation, a Hawkes model that does not consider leading-triggered non-stationarity could overestimate. Finally, we also designed and implemented a quantitative pair trader, utilizing the excitation influence of te leader sector on the audience sector, which achieved remarkable excess returns comparing with the benchmark.

An important limitation of our model is that, as it was presented, it treats equally all the MA crossover bursts. If there is an extremely great MA crossover bursting in the estimation interval, and this burst has different characteristics that lead to very different impacts, the model is expected to average the differences or to be influenced by the most relevant ones. Today, machine learning and deep learning is very popular in stock forecasting, [40,41] both have successfully applied machine learning methods for stock predictability and achieved fantastical and valuable results. Our model may also provide potential machine learning features for stock predictability. For example, according to our model, one can use the number of crossovers, the intensity of excitation, the duration of excitation, etc. to extend the features pool.

Despite being focused on financial market data, the approach presented in this study could be useful in the analysis and modeling of other complex systems. In fact, the presence of exogenous and endogenous drivers of the activity is ubiquitous in other systems monitored in continuous time. Our model can be possibly applied into bond convergence trading. Actually, convergence trading is a trading strategy consisting of two positions: buying one asset forward and selling a similar asset forward (going short the asset) for a higher price. For example, we can sell the 30-year US treasury bond short and buy the 29½-year US treasury bond. Compared with stocks, we can regard the rise in bond price to a certain level as a golden crossover and the decline in bond price to a certain level as a death crossover. There may also be some mutual excitation mechanism between the golden crossovers or the death crossovers of different bonds. The 30-year bonds generally trade at a premium over the 29½-year bonds, because they are more liquid—there is a liquidity premium. Once a newer bond is issued, this liquidity premium will generally decrease or disappear. Therefore, the rise of the 29½-year US treasury bond prices (golden crossover) may excite the decline in the 30-year US treasury bond prices (death crossover). Actually, our model can be modified to figure out the impact of death crossovers’ arrivals within one bond on golden crossovers in another bond. Describing the different source of excitation as correlated point processes is in general quite complicated, and the Hawkes approach proposed here could be a useful method to model, fit, and evaluate the leading relationship between the several drivers of generic stochastic dynamics.

## Figures and Tables

**Figure 1 entropy-23-01411-f001:**
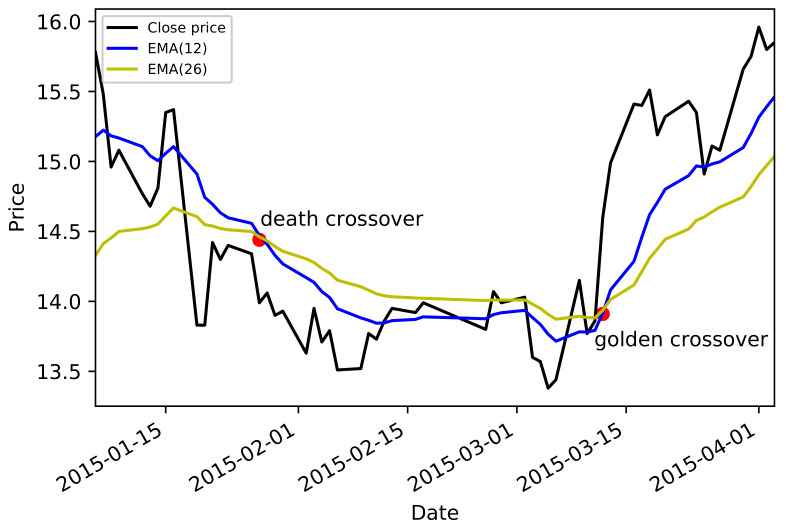
(Color online) EMA golden and death crossover. Red points identify golden and death crossovers of an individual stock 000001.XSHE during the period: (20150106, 20150405). Blue line: short-term EMA with $k_{s}=12$ days. Yellow line: long-term EMA with $k_{l}=26$ days. Black line: daily close price line.

**Figure 2 entropy-23-01411-f002:**
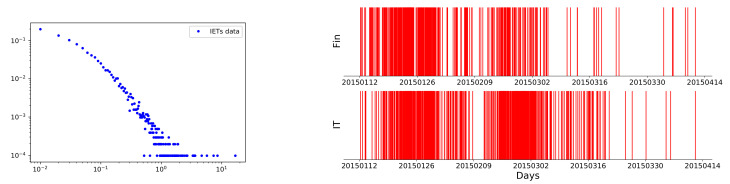
**Left panel**: (color online) the distribution of inter-event times. This figure refers to the data of the financial sector. **Right panel**: (color online) demonstration of first EMA golden crossovers. Each red line represents a crossover event happening at a certain time along the horizontal axis. Upper panel: first EMA golden crossover events of each stock in the financial sector during the period (20150112,20150414]). Lower panel: the case in IT sector during the same period.

**Figure 3 entropy-23-01411-f003:**
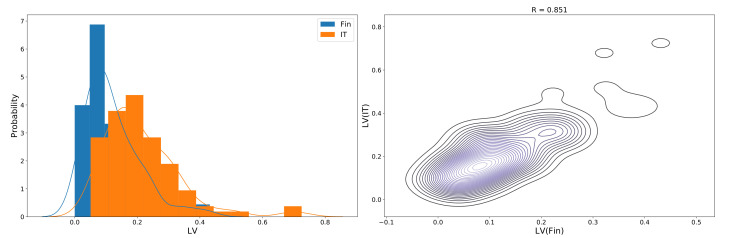
(Color online) **left panel**: distribution of two LV sequences in thenumber time windows. The financial sector data are plotted in blue, while the IT sector data are in orange. **Right panel**: Pearson correlation coefficient between the IT and financial sectors.

**Figure 4 entropy-23-01411-f004:**
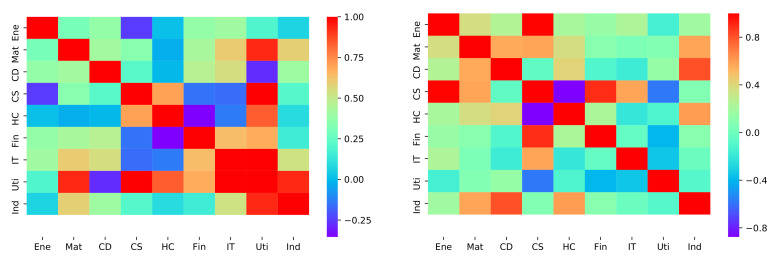
The LV’s heat map of golden and death crossovers. **Left panel**: golden crossover **Right panel**: death crossover.

**Figure 5 entropy-23-01411-f005:**
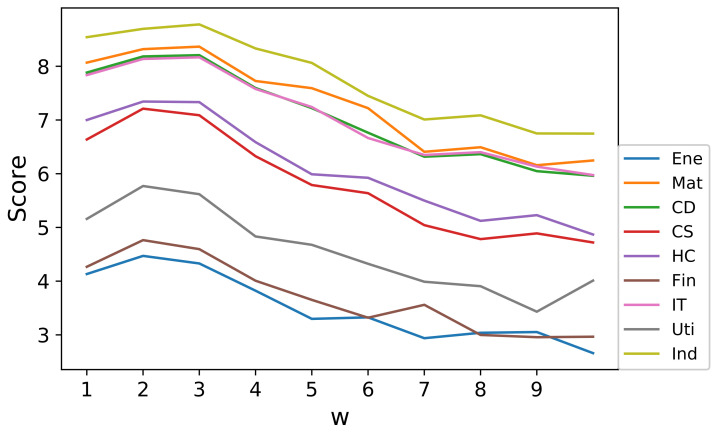
Score function for different values of w and different sectors.

**Figure 6 entropy-23-01411-f006:**
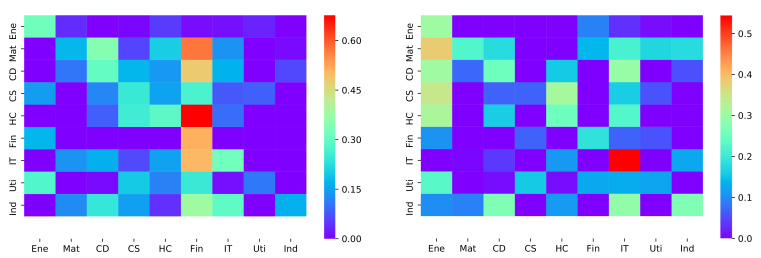
The heat map of golden and death crossovers. **Left panel**: golden crossover. **Right panel**: Death crossover.

**Figure 7 entropy-23-01411-f007:**
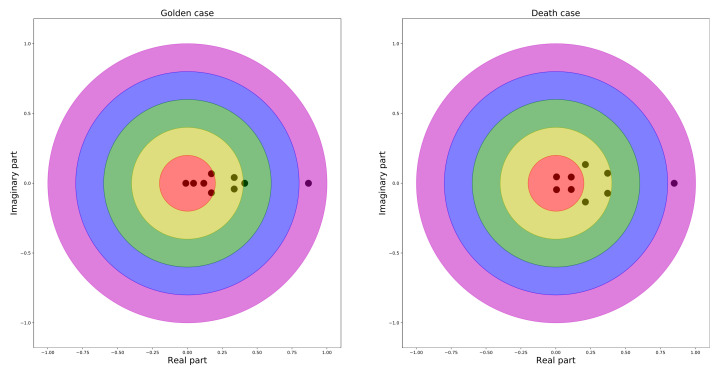
The spectral circle of the eigenvalues of both golden and death crossover matrices. **Left panel**: golden crossover. **Right panel**: death crossover.

**Figure 8 entropy-23-01411-f008:**
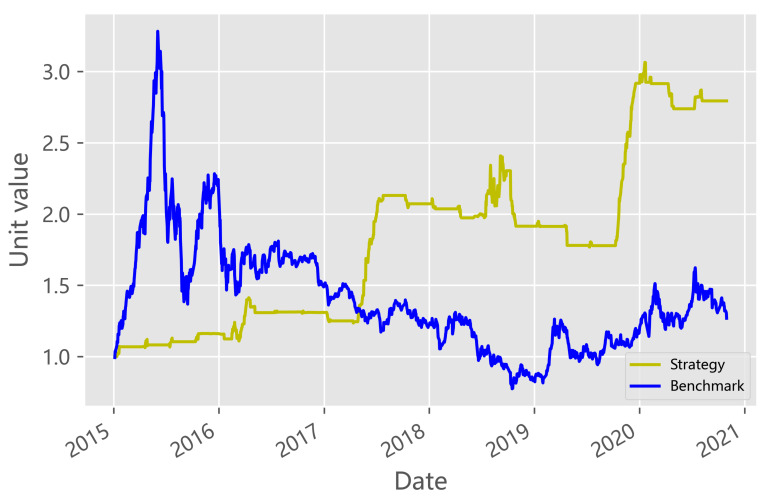
Daily return of the two strategies over the assetment period.

**Table 1 entropy-23-01411-t001:** Some statistical results of stock sectors. The names listed in the first line are the abbreviations of energy, materials, consumer discretionary, consumer staples, health care, financials, information technology, telecommunication services, utilities, and industrials respectively.

Sectors	Ene	Mat	CD	CS	HC	Fin	IT	Tel	Uti	Ind
No. of stocks	83	618	596	214	293	98	668	4	121	924
No. of golden crossovers	847	5641	5274	2244	2813	978	5648	40	1384	8463
No. of death crossovers	812	5355	5011	2148	2671	920	5388	39	1354	8164
Avg. interval of golden crossover	13.9	12.8	11.6	12.4	12.8	11.1	12	37.4	11.4	10.1
Avg. interval of death crossover	14.2	13.8	12.6	13.4	13.4	11	12.5	36.9	11.2	12.9

**Table 2 entropy-23-01411-t002:** The power-law constants obtained by fitting the IET distributions for each sector.

Ene	Mat	CD	CS	HC	Fin	IT	Uti	Ind
2.3734	2.8053	2.6929	3.0038	2.6672	2.0956	2.7666	2.3319	2.5408

**Table 3 entropy-23-01411-t003:** Optimal parameters for each sector.

	**Golden Crossover**								
**Sectors**	**Ene**	**Mat**	**CD**	**CS**	**HC**	**Fin**	**IT**	**Uti**	**Ind**
μ	0.317	1.509	1.252	0.700	0.709	0.303	1.498	0.495	2.092
α	0.512	0.609	0.709	0.667	0.720	0.595	0.661	0.590	0.619
β	0.544	0.707	0.712	0.692	0.730	0.634	0.696	0.607	0.632
*n*	0.941	0.861	0.996	0.964	0.986	0.938	0.950	0.972	0.979
	**Death Crossover**								
**Sectors**	**Ene**	**Mat**	**CD**	**CS**	**HC**	**Fin**	**IT**	**Uti**	**Ind**
μ	0.349	1.328	1.371	0.728	0.847	0.323	1.274	0.626	1.858
α	0.351	0.648	0.616	0.614	0.614	0.469	0.679	0.492	0.631
β	0.366	0.649	0.669	0.637	0.653	0.482	0.711	0.516	0.666
*n*	0.959	0.998	0.921	0.963	0.940	0.973	0.955	0.953	0.947

**Table 4 entropy-23-01411-t004:** The $\mu^{i}$ values of univariate model and multidimensional model together with the average number (activity) of golden and death crossover events in each sector.

	**Golden Crossover**								
**Sectors**	**Ene**	**Mat**	**CD**	**CS**	**HC**	**Fin**	**IT**	**Uti**	**Ind**
Activity (events/day)	0.741	4.897	4.559	1.927	2.388	0.834	4.833	1.204	7.317
μ	0.029	0.69	0.428	0.013	0.001	0.232	0.858	0	2.092
Single μ	0.317	1.509	1.252	0.7	0.709	0.303	1.498	0.495	2.092
	**Death Crossover**								
**Sectors**	**Ene**	**Mat**	**CD**	**CS**	**HC**	**Fin**	**IT**	**Uti**	**Ind**
Activity (events/day)	0.726	4.692	4.369	1.868	2.273	0.791	4.617	1.195	7.14
μ	0.165	0.208	0.52	0.193	0.342	0.032	0.686	0.019	0.77
Single μ	0.349	1.328	1.371	0.728	0.847	0.323	1.274	0.626	1.858

**Table 5 entropy-23-01411-t005:** The average impact of each sector on the other ones.

Sectors	Ene	Mat	CD	CS	HC	Fin	IT	Uti	Ind
Golden case mean	0.101	0.064	0.141	0.128	0.126	0.405	0.125	0.025	0.026
Death case mean	0.234	0.044	0.11	0.033	0.107	0.061	0.219	0.047	0.07

**Table 6 entropy-23-01411-t006:** The details of strategy.

Signal	Operation
The number of golden EMA crossovers of the leading sector within one day exceeds a certain threshold $\Delta_{B}$	Buy stocks without any EMA golden crossovers when the signal arrives
The number of death EMA crossovers of the leading sector within one day exceeds a certain threshold $\Delta_{S}$	Sell stocks in long position without any EMA death crossovers when the signal arrives
MACD of the stock in long position is smaller than 0	Sell the corresponding stock

## Data Availability

3rd Party Data Restrictions apply to the availability of these data. Data was obtained from [Tushare] and are available http://tushare.org/ (accessed on 20 October 2021) with the permission of [Tushare].

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
