# Peer review of "Detection of Mutual Exciting Structure in Stock Price Trend Dynamics"

_entropy, 2021, doi:10.3390/e23111411_

Round 1

Reviewer 1 Report

"Detection of mutual exciting structure in stock price trend
dynamics"
by Shangzhe Li, Xin Jiang, Junran Wu, Lin Tong, Ke Xu 

The authors of this paper studied the dynamics of stock price using the Hawkes model to capture mutual exciting activities of both endogenous and exogenous factors. The authors used the three-year data of 9 sectors in Chinese markets. To this end they measured the inter-event times of the DMAC (Dual Moving Average Crossovers) and the correlation of local variation (LV).  The necessity is to suppose a kernel (Eq. 6) for each sector and construct an equation describing the intensity of arrival events (Eq. 5). The mutual model which combines local trends and intersector trends is given by Eq. 7. Among the most important results, one can mention the case of multiple sectors which is concretely shown in Fig. 8 where one sees that the strategy which consists in supposing a leader sector dominating an audience sector generates interesting shift in daily asset return. But the authors have introduced another indicator MACD to avoid the fluctuation risk to get that result.

Here are my remarks on the paper:
1. There are too many abbreviations which are not necessary. These make the reading very hard.  
2. The presentation of the model is lengthy. I am not sure about the assumption in section 2.2 where the authors take only the first crossover in a given sector. For me what matters is the outcome of the day if we take the day as the time basis, not the first crossover in the morning for example. There are too many assumptions in the model making it hard to use. The authors explained the reason for each assumption but I have the feeling that too many are introduced by hand for convenience. 
3. On the same idea of the combination of effects due to intra-stock and inter-stock interactions, there is a paper in Physica A 570 (2021) 125813 which studied the dynamics of the price behavior in stock markets by a statistical physics approach.  The model is defined once for all and among the consequences stemming from that one finds the persistent effect similar to Fig. 8.  I recommend the authors to look at this paper and make some comparative comments.

I believe that this paper can be published after some revisions to take into account the above remarks.  

Reviewer 2 Report

This paper looks at an interesting aspect of mutual exciting structure in financial market (especially stock market trend). I am just have three points: 

  1. The literature about its applications should be elaborated. Please mention how your research outcomes could be incorporated with the machine learning or deep learning model for stock predictability (see at [a] and [b])

[a] Abdullah, M. (2021), "The implication of machine learning for financial solvency prediction: an empirical analysis on public listed companies of Bangladesh", Journal of Asian Business and Economic Studies, Vol. ahead-of-print No. ahead-of-print. https://doi.org/10.1108/JABES-11-2020-0128

[b] Zhao, Y. and Chen, Z. (2021), "Forecasting stock price movement: new evidence from a novel hybrid deep learning model", Journal of Asian Business and Economic Studies, Vol. ahead-of-print No. ahead-of-print. https://doi.org/10.1108/JABES-05-2021-0061 

2. Please also elaborate the strategy in meeting death crossover and golden crossover. How should we do at the wise decision?

Overall, I would like to accept this paper without any objection.

Reviewer 3 Report

Interesting paper and reasonably written with clear explanations.

Needs another pass at checking the English

[1] On page 5 please give more details on the IET distributions being of the power law form e.g. what were the power law constants for the assets and what are the implications for the variances of these distributions

[2] Could you apply this framework for bond convergence trading? Please discuss how the methods in this paper would have to be modified for this.

Round 2

Reviewer 3 Report

I am satisfied with the author responses.